# Feasibility of Encord Artificial Intelligence Annotation of Arterial Duplex Ultrasound Images

**DOI:** 10.3390/diagnostics14010046

**Published:** 2023-12-25

**Authors:** Tiffany R. Bellomo, Guillaume Goudot, Srihari K. Lella, Eric Landau, Natalie Sumetsky, Nikolaos Zacharias, Chanel Fischetti, Anahita Dua

**Affiliations:** 1Division of Vascular and Endovascular Surgery, Massachusetts General Hospital, Boston, MA 02114, USA; guillaume.goudot@aphp.fr (G.G.); srihari.lella@mgh.harvard.edu (S.K.L.); sumetsky@gmail.com (N.S.); nzacharias@mgh.harvard.edu (N.Z.); adua1@mgh.harvard.edu (A.D.); 2Harvard Medical School, Massachusetts General Hospital, Boston, MA 02114, USA; cfischetti@bwh.harvard.edu; 3Encord, Cord Technologies Inc., New York City, NY 10013, USA; eric@encord.com; 4Department of Emergency Medicine, Brigham and Women’s Hospital, Boston, MA 02115, USA

**Keywords:** deep learning, artificial intelligence, popliteal artery aneurysm, ultrasound

## Abstract

DUS measurements for popliteal artery aneurysms (PAAs) specifically can be time-consuming, error-prone, and operator-dependent. To eliminate this subjectivity and provide efficient segmentation, we applied artificial intelligence (AI) to accurately delineate inner and outer lumen on DUS. DUS images were selected from a cohort of patients with PAAs from a multi-institutional platform. Encord is an easy-to-use, readily available online AI platform that was used to segment both the inner lumen and outer lumen of the PAA on DUS images. A model trained on 20 images and tested on 80 images had a mean Average Precision of 0.85 for the outer polygon and 0.23 for the inner polygon. The outer polygon had a higher recall score than precision score at 0.90 and 0.85, respectively. The inner polygon had a score of 0.25 for both precision and recall. The outer polygon false-negative rate was the lowest in images with the least amount of blur. This study demonstrates the feasibility of using the widely available Encord AI platform to identify standard features of PAAs that are critical for operative decision making.

## 1. Introduction

Lower extremity vascular disease is the third leading cause of atherosclerotic morbidity [1], affecting 7%, or 8.5 million, of the adults in the United States [2]. For 2% of these affected patients, the end stage of this disease results in amputation [3]. There are even higher rates of amputation in specific subsets of lower extremity atherosclerotic diseases: popliteal artery aneurysms (PAAs) result in a 15% amputation rate [4] and account for 70% of all peripheral arterial aneurysms [5]. Given the high disease burden and amputation rate, routine surveillance is recommended at least annually. The method of choice for the surveillance of lower extremity vascular disease, including PAAs, is duplex ultrasound (DUS) [6]. PAA DUS provides surveillance by reporting measurements relevant for operative decision making, including diameter and the existence of a mural thrombus [7].

Ultrasound (US) has become an indispensable tool for vascular clinical practice as a low-cost, no-radiation, real-time dynamic imaging display of vasculature [8]. Despite these advantages, US image quality can be adversely affected by operator acquisition and noise, such as bone shadowing [9]. DUS velocity measurements for stenosis are often distorted by echogenic mismatch of the blood and vessel wall [10]. To measure the vessel wall and the true lumen for an accurate percent of stenosis, manual segmentation has been applied, although it is not routinely performed due to the large amount of time, effort, and individual variability these measurements incur [11]. To eliminate this subjectivity and provide efficient segmentation, artificial intelligence (AI) has been applied to accurately delineate many different structures on US.

AI and machine learning algorithms have been increasingly utilized to aid in the detection [12], quantification, and even diagnosis [13] of diseases based on medical imaging [14]. The application of these AI methodologies to US images in particular has resulted in great advances for many diseases: AI models for breast cancer detection have over 80% sensitivity and specificity [15,16]. Thyroid nodules detected on US have also been accurately segmented and classified [17] using convoluted neural networks (CNNs) without user interaction [18]. Similarly, CNNs followed by unsupervised clustering have been used to identify vasculature among hepatic tissue in liver US with 70% accuracy [19]. AI applied to US imaging for cardiovascular diseases has largely focused on echocardiograms [20] and intravascular coronary US [21]. The successful application of this technique has distinguished between layers of coronary arteries that are of similar echogenicity to define stenosis [22]. This same technique of distinguishing vessel wall layers has been applied to carotid US: CNN [14,23] and deep learning [8,18] models have been developed to identify, segment, and quantify carotid intima–media thickness on US images.

Carotid US is the only type of vascular US imaging with robust AI models. However, there is an opportunity to expand these AI models to all types of vascular US performed for follow-up surveillance. For PAAs specifically, DUS is performed on at least a yearly basis, as dictated by the Society for Vascular Surgery (SVS), to identify features of popliteal arteries that dictate the need for repair [7]. There is no standard reporting guideline for potentially concerning clinical features. However, there are recommended features that are associated with thromboembolism, including diameter, patent channel area, and percent stenosis [24]. RPVI-certified specialist annotation of features can be time-consuming, error-prone, and operator-dependent [25]. Automatic segmentation by AI can reduce dependencies on operators and aid with the standardization of time-consuming PAA measurements. In addition, training AI with multiple different US images, including B-mode grayscale images and Doppler color images, could also improve the diagnostic accuracy of models [15]. Encord (Cord Technologies Limited, London, United Kingdom) is an easy-to-use, readily available online computer vision platform designed for automating and managing annotations for medical AI applications. Encord has a subsection specifically dedicated to medical segmentation, previously used by King’s College London for automated polyp detection and by the Stanford Department of Medicine for the segmentation of cells in microscopy images for lung B line identification. However, there are no current published research studies on the use of the Encord platform in medical segmentation for ultrasound. In this study, we tested the feasibility of the Encord platform to create an automated model that segments the inner lumen and outer lumen within PAA DUS. Using machine learning segmentation to identify the maximal diameter and thrombus area within PAAs will help standardize DUS measurements that are critical for operative decision making.

## 2. Materials and Methods

This cohort was derived from a previously collected data set of 28 patients with PAAs. Briefly, the cohort was derived from the Massachusetts General Brigham (MGB) Research Patient Data Registry (RPDR), a multi-institutional repository that gathers demographics, diagnosis codes, encounter data, procedural codes, medications, and other patient clinical information. The RPDR database was queried for all patients with a diagnosis code for “Aneurysm of the artery of lower extremity” (ICD9 442.3/ICD10 I72.4) and a pre-operative DUS from inception of the database in 2008 to 2022. Manual chart review was then performed to confirm the presence of a PAA. The Partners Human Research Committee Institutional Review Board approved this study protocol for patients >18 years of age and patient consent to participate was waived (IRB # 2019P003163).

Of the DUS performed for PAA surveillance pre-operatively, 100 of the highest-quality cross-sectional images were selected by Registered Physician in Vascular Interpretation (RPVI)-certified co-authors (G.G., A.D.). In total, 65% of the images selected were B-mode images and the other 35% were Doppler images, as there is prior evidence to support training models of both B-mode and Doppler images to improve the diagnostic accuracy of models. The 100 images were selected from 100 separate DUS encounters for PAA pre-operative surveillance across a total of 44 patients. The 100 images were deidentified and extracted as PNG images for upload to the Encord annotation platform.

A Data Use Agreement and a Data Transfer Agreement were made between the General Hospital Corporation with MGB and Cord Technologies Inc with Encord for use of deidentified DUS images. The Encord platform offers online annotation tools specifically for medical-grade radiology image segmentation. Ontology is defined as a single measurement a model could be trained to perform, and Encord allows several ontologies to be created simultaneously on a single image. These ontologies can then be used to train the proprietary Encord AI technology for automated segmentation on sparse amounts of data. These proprietary micro-models are based on the Mask Region-Based Convolutional Neural Networks (Mask-RCNN) model architecture implemented in the PyTorch framework.

After uploading the 100 deidentified images to Encord, RPVI-certified co-authors (G.G., A.D.) both used Encord’s segmentation tools to manually segment both the inner lumen and outer lumen of the PAA. The outer lumen was defined as the outer boundaries of the PAA measured by creating a polygon around the outermost wall of the vessel pictured (Figure 1A). The inner lumen was defined as the patent channel area where blood flows and was measured by creating a polygon around the innermost wall of the vessel pictured (Figure 1C). This segmentation captures not only the largest diameter of the PAA, but also a measure of thrombus by understanding the difference between the inner and outer lumen. Each co-author segmented and validated each image. When a discrepancy was identified, both authors discussed with a third-party RPVI-certified co-author (N.Z.) to resolve the disagreement. After manually segmenting and validating the inner and outer polygons for each image to be used as ground truth, three different segmentation models were created using the proprietary AI technology on Encord’s platform. The models all employ a Mask R-CNN backbone trained over 500 epochs on the ultrasound data set (Appendix A). The first model was trained on 20 images and tested on 80 images, the second model was trained on 60 images and tested on 40 images, and the last model was trained on 20 images.

The Encord platform is designed with features to identify model accuracy. Encord uses Intersection over Union to evaluate model performance, where geometric shapes are quantified to determine the intersection of these two shapes. Intersection over union was calculated by quantifying the area of overlap between the AI-model-created polygon and the ground-truth-defined polygon and then dividing by total area captured by both the AI-model-created polygon and the ground-truth-defined polygon. A true-positive was defined as a 50% match between the union and intersection. A false-positive was defined as less than 50% match between the union and intersection. A 50% match was chosen as the cutoff for accurate segmentation based on prior literature data [26]. Mean Average Precision (mAP) was defined as the average of the precision scores at different thresholds of Intersection over Union (IoU). Precision was computed by evaluating the overlap between predicted segments and ground-truth segments at various IoU thresholds, and the Average Precision (AP) was calculated. mAP was then the mean of these AP values, providing a single metric to evaluate the performance of the segmentation algorithm. Blur was defined as a metric that quantifies the variance of the output of a Laplacian filter when applied to an image. Precision was defined as the ratio of true-positive predictions to the total number of positive predictions, both true and false. Recall was defined as the ratio of true-positive predictions to the total number of actual positives, including both true-positive and false-negative predictions. An embedding plot was created by the Encord platform, which is a means to translate deep learning into a coordinate system for conceptualization.

## 3. Results

There were 100 DUSs among 44 patients with demographics listed in Table 1.

The patients were 100% male, 95% white, and a median of 76 years old. Most patients had a right-sided PAA at 75%. In terms of vascular risk factors, most patients were ever smokers at 61%, had hyperlipidemia at 91%, and hypertension at 89%. We trained three different models with increasing amounts of images and tested the models with the remaining subset of images. An example of a true-positive and false-positive for both the inner and outer polygon within our model is shown in Figure 1.

The mean Average Precision (mAP) for object detection of the outer polygon was 0.85 for the 20-image model, 0.06 for the 60-image model, and 0 for the 80-image model (Table 2).

The mAP of the inner polygon was 0.23 for the 20-image, 60-image, and 80-image models (Table 2). The true-positive rate (TPR) for the inner polygon remained 0.23, whereas the TPR for the outer polygon differed with training models: the 20-image model had a TPR of 0.86, the 60-image model had a TPR of 0.18, and the 80-image model had a TPR of 0 (Table 2).

Our best-performing model was trained on 20 images and tested on 60 images, where the model for the outer polygon incorrectly identified 3 out of 22 clinician-labeled DUSs and the model for the inner polygon incorrectly identified 17 out of 22 clinician-labeled DUSs (Table 3).

The precision and recall of the model for the inner polygon, outer polygon, and average overall model performance were quantified and are graphed in Figure 2.

The average overall model had a score of 0.50 for both precision and recall. The outer polygon had a higher recall score than precision score at 0.90 and 0.85, respectively. The inner polygon had a score of 0.25 for both precision and recall.

To further explore the performance of the model in relation to blur, we plotted precision on the *y*-axis and the blur metric on the *x*-axis using the Encord platform (Figure 3).

Most of the outer polygon had a blur metric of negative 200 with an average precision of 0.85 on the *y*-axis. Most of the inner polygon also had a blur metric of negative 200 with an average precision of 0.82 on the *y*-axis. Conversely, we plotted the false-negative rate on the *y*-axis and the blur metric on the *x*-axis (Figure 4).

The outer polygon false-negative rate was the lowest in images with the least amount of blur and had an average false-negative rate of 0.18. The inner polygon false-negative rate was also lowest in images with the least amount of blur. However, there was wide variability, causing the average false-negative rate to be much higher at 0.79. The first 20 images had an average blur of −138 in comparison to an average blur of −183 for the entire data set. (Table 4). Doppler images were present at around 35% for all image models.

## 4. Discussion

In this study, we tested the feasibility of using a widely available online computer vision platform, Encord, to create an automated model that segments both the inner lumen and outer lumen in PAA DUS images. We identified 100 images among 44 patients with manually confirmed PAAs. We uploaded these to the Encord platform, used Encord annotation tools to manually segment the inner and outer lumen as ground truth, and trained models on increasing subsets of images. We subsequently analyzed model performance on a variety of metrics, including mAP, recall, and TPR. The best-trained model on the smallest training set segmented the outer lumen of PAAs with good precision and accuracy, demonstrating the feasibility of using Encord to identify the standard features of PAAs that are critical for operative decision making.

The generation of the models discussed proved highly feasible from technical, operational, and results perspectives. No specialized or technical training was needed to utilize the Encord platform for model training or testing. We uploaded our ultrasound files to the Encord platform as a data set. We then effortlessly attached this data set to a project. Within the project, we created two ontologies for inner and outer lumen segmentation. The operationalization of the model training system was seamless, facilitated by the Encord platform’s capability to allow two authors to independently segment each image and verify segmentation within the platform. Discrepancies identified by the platform were promptly flagged for easy identification. Following the annotation and validation of images with the agreed-upon ground truth, model training parameters—specifically, the Mask R-CNN backbone and 500 epochs—were selected within the Encord platform. After training models on subsets of images, testing was also conducted within the Encord platform. The desired tests were simply selected in the analysis tab of the project and, after a runtime period, the platform presented calculations of true-positives, false-negatives, mAP, IoU, and blur. In summary, the platform’s intuitive navigation, complemented by tutorials for both model training and analysis, allowed for straightforward operationalization of the model training system among members of the research team. The results were displayed in an understandable format and interpreted within the following discussion.

Counterintuitively, we found that the best-performing models were trained on the smallest data set of 20 images. The smaller data sets had better precision and accuracy than the larger data sets, possibly due to image features and quality. One component of image quality often quantified in AI training is blur, a phenomenon where the details of an image are not clearly visible and sharp edges become smooth [27]. Intuitively, our analysis found that as the images become blurrier, the model precision declined, and false-negative rates increased. There have been many sophisticated algorithms developed to quantify [28] but also remove these images from training models [29,30]. Removing blur from [31] or augmenting blur [27] in images can be important for training accurate AI models [32]. However, the quality of image acquisition depends on many factors that have been areas of advancement in recent years [33]. Easily modifiable factors include gain levels on the machine, the ultrasound gel used, the amount of probe contact with the patient, and ensuring the patient remains still [34]. There are also difficult-to-modify factors in obtaining high-quality images, including the quality of the machine used and the skill of the technologist who acquired the image [35]. Some factors for image quality are unmodifiable, including an individual’s body habitus, scar tissue from prior surgeries, and overall inflammation in tissues [36,37]. All of these factors and human error must be taken into consideration if AI is used to segment medical images of varying quality, and it may be important to train a model on a wider spectrum of quality. On further analysis in this study, the first 20 images used to train the model have some high contrast features and a low blur metric in comparison to other images, which could explain in part why increased training data did not improve the model. The high contrast features used in the first 20 images included some color Doppler, where a red color indicates forward flow towards the US probe and a blue color indicates flow away from the US probe. Seven out of the twenty images used to train the best-performing models included color Doppler, and there is evidence to support that training AI with color Doppler images improved model performance [15]. Although color Doppler directionality changes depending on the position of the probe, the Encord platform does not include predefined color scales or gradient thresholds. This absence raises the prospect that a model may acquire the ability to identify both blue and red colors as equivalent entities with sufficient training. Our models may have identified both blue and red colors to be the inner segmentation of an open lumen, which is a desired interpretation within the context of this study.

With regard to the models for outer and inner polygons, the outer polygon model outperformed the inner polygon model on every metric. The outer polygon demonstrated almost equal precision and recall at 0.85. The mAP for the outer polygon model was 0.85 with a true-positive rate of 0.86, which is comparable to other clinically used high-performing models for US segmentation. Thyroid nodule identification on US is a highly developed area, where CNN modes achieve a true-positive rate of 0.95 for identification. Femoral nerve segmentation models trained on a subset of 50 ultrasound images also performed well with an accuracy of 84% [26]. Specific to vessel segmentation, the only robust models developed focus on measuring carotid intima media thickness, which involves segmenting different layers of the arterial wall [18]. One CNN model using 503 images derived from 153 patients achieved a classification performance of 89% sensitivity and 88% specificity for carotid intima media thickness [23]. In this study, the model developed for segmentation of the inner PAA lumen, however, performed poorly, with an mAP of 0.23 and a true-positive rate of 0.23. This low mAP and low PTR for inner PAA lumen segmentation likely reflects the model’s inability to differentiate between the inner wall and adjacent thrombus, as these adjacent structures are of similar echogenicity on a cross-sectional US image [19]. The task of circumferentially segmenting an inner patent lumen in this study poses more difficulty than measuring the thickness of a single linear layer in a longitudinal view [23].

In terms of overall clinical application, the field of ultrasound (US) has presented substantial opportunities for the integration of AI. Inherent subjective characteristics of US can be improved with the integration of AI, including grayscale imaging quality, which is adversely affected by operator acquisition [9], and noise in relationship to other structures [10]. The clinical need for segmentation in US has been substantially advanced by AI technology, such as in breast cancer detection [15,16], thyroid nodule classification [17,18], and hepatic tissue vasculature identification in liver US [19]. Specifically, within cardiovascular disease, AI has been successfully used to aid with accurate segmentation of the four chambers of the heart [38]. Heart morphology can be affected by disease factors, causing wall thickening, remodeling, and pressure changes that are difficult to manually collect for each image and are subjective based on the experience of both the technician and the interpreter. Technology has been developed to automatically segment 2D or 3D images of the heart, where automatic and accurate measurements of cardiac cavity size can be performed. The benefits of automation include not only time but also accuracy: in a convolutional neural network model trained on 14,000 images, automated measurements were either comparable or superior to manual measurements of cardiac chambers and ejection fraction [39].

The same challenges surmounted by AI within cardiovascular US currently persist in the field of vascular US [14]. The quality of each vascular image is also based on the experience of the technician, and the final report generated is based on the subjectiveness of the interpreter. The Intersocietal Accreditation Commission (IAC) has initiated standards for vascular US acquisition and reports, but not every center is IAC-accredited. In addition, the IAC has no current image acquisition or result reporting standards for PAA DUS, and therefore image protocols are left up to the internal protocols of each institution [24,40]. The SVS guidelines focus mainly on quantifying and reporting PAA size [7]. However, size alone does not singularly dictate the need for operative repair; studies have demonstrated that thrombus burden and the percent of thrombus also portends a high risk of thromboembolic events and amputation [4,40,41,42]. Manual segmentation of the vessel lumen to identify these high-risk features is difficult given the similar echogenicity of adjacent plaques. This similar problem in carotid US has been resolved with the use of AI: machine learning has been applied to the measurement of carotid artery intima–media thickness [23], segmentation of the vascular lumen [8], and classification of carotid vascular plaque components [43]. In this study, we were able to train an existing easy-to-use AI platform on the identification and segmentation of the vascular inner and outer lumen. These measurements can be used to abstract a diameter and percent thrombosis, which are high-risk features of PAA resulting in thromboembolic events [7]. Clinically, applying a model to PAA US has the potential to eliminate measurement subjectivity and provide efficient segmentation for result reporting. The ideal real-world application would include uploading all PAA DUS images to the Encord platform for segmentation and calculation of the largest diameter of PAA and the highest percent of thrombus burden within the collection of images. Although the clinical application at this phase in development is severely limited, this study provides a foundation for the creation of a more robust AI model.

This feasibility study has several large limitations. The image format used in this study was PNG, which is a lower-quality image than Digital Imaging and Communications in Medicine (DICOM) images, which are the international standard. The lower-quality imaging limits the generalizability of this study. The Encord platform is equipped to handle DICOM storage and future studies should aim to use medical-grade DICOM images when training AI models. We found the best-performing model was trained on a small data set of 20 images, which could also represent overfitting of the data that struggles to perform well with new data. A larger number of high-quality popliteal artery ultrasound images should be used to train AI models. In addition, training a model with a large number of both higher- and lower-quality images could improve the generalizability of a model. In this study, the selection of images by RPVI-certified physicians introduced bias and decreased the generalizability of this model to all images captured. However, there remains ample opportunity to expand the concept of AI in vascular ultrasound to abdominal aneurysm identification of inner and outer lumen. This study limited segmentation to outer and inner lumen, as this ontology was the simplest for the Encord platform to handle. Future studies should include other dimensions, including surface area, length, and width. We did not include any normal arteries in the training model, which may be easier training material for AI and could be used in future model creation. This study specifically focused on using AI developed by Encord as a feasibility test. However, Encord’s generated models could be compared to other AI-generated models for accuracy and performance. Providing the same images and comparing the models developed would be an informative study to determine the advantages and disadvantages of each platform with regard to vascular ultrasound.

## 5. Conclusions

We report the use of a widely available Encord AI platform to develop the first automated model for segmentation of both the inner and outer lumen in PAA DUS images. This study demonstrates the feasibility of using Encord to identify the standard features of PAAs that are critical for operative decision making. Using AI to automatically segment features of PAA that are of clinical interest has the potential to improve efficiency, eliminate operator subjectivity, and provide a set of standardized PAA characteristics for clinical decision making.

## Figures and Tables

**Figure 1 diagnostics-14-00046-f001:**
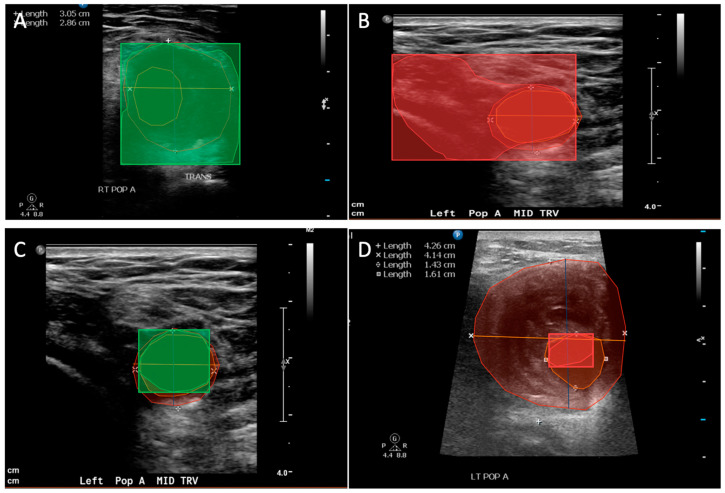
AI segmentation classifications on duplex ultrasound images. (**A**) Outer polygon true-positive classification, where the color green indicates a correct segmentation. (**B**) Outer polygon false-positive classification, where red indicates an incorrect segmentation. (**C**) Inner polygon true-positive classification, where the color green indicates a correct segmentation. (**D**) Inner polygon false-positive classification, where red indicates an incorrect segmentation.

**Figure 2 diagnostics-14-00046-f002:**
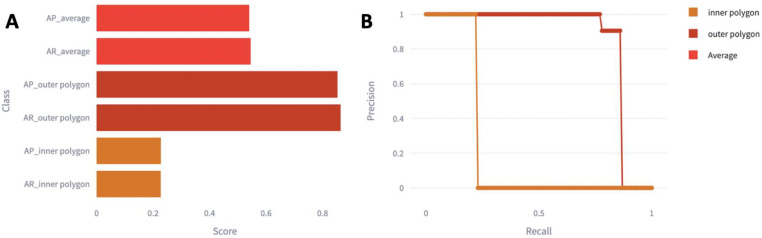
Precision and recall scores for all models. (**A**) The average precision and recall scores for each model, where 0 is poor and 1 is perfect precision and recall. (**B**) Precision was plotted in relation to recall for a precision–recall curve. Abbreviations: AP, Average Precision; AR, Average Recall.

**Figure 3 diagnostics-14-00046-f003:**
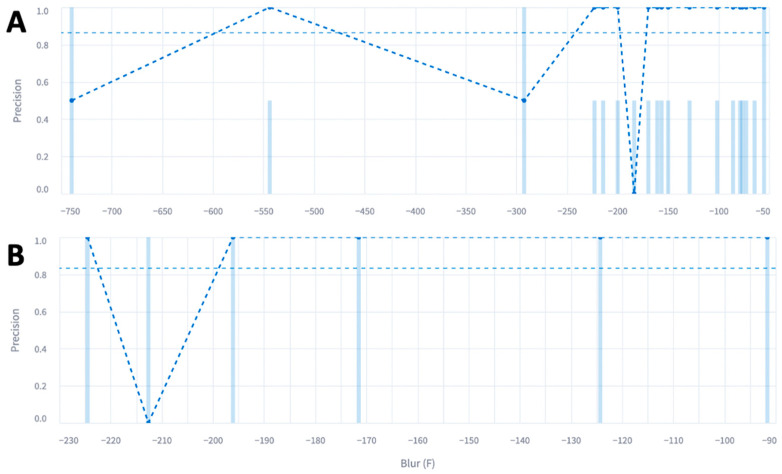
Precision of the (**A**) outer and (**B**) inner polygons with respect to the blur metric. Each straight vertical blue line represents an image and the blue dot represents the precision for that cluster of images. The dotted straight horizontal line represents the average precision across all images.

**Figure 4 diagnostics-14-00046-f004:**
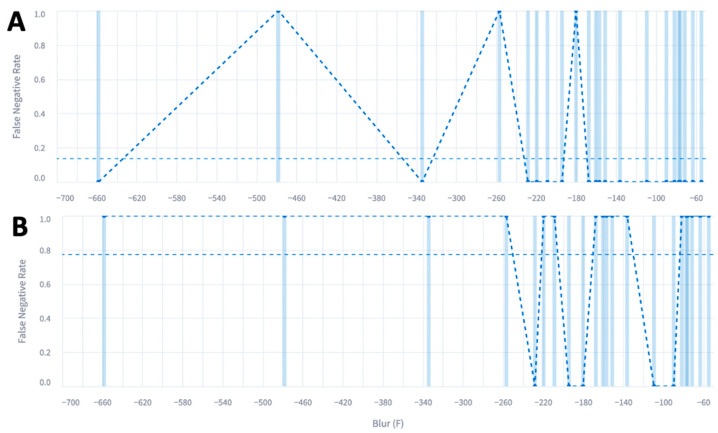
False-negative rate of the (**A**) outer and (**B**) inner polygons with respect to the blur metric. Each straight vertical blue line represents an image and the blue dot represents the false negative rate for that cluster of images. The dotted straight horizontal line represents the average ffalse negative rate across all images.

**Table 1 diagnostics-14-00046-t001:** Characteristics and demographics documented per individual patient (*n* = 44 patients).

	Total Number of Patients
Race *n*(%)	
White	42 (95)
African American	2 (5)
Sex *n*(%)	
Male	44 (100)
Age (median years (IQR))	76 (56, 93)
Laterality of PAA *n*(%)	
Left *n* (%)	11 (25)
Right *n* (%)	33 (75)
Ever Smoker	27 (61)
Hyperlipidemia	40 (91)
Hypertension	39 (89)
Type 2 Diabetes	13 (30)

**Table 2 diagnostics-14-00046-t002:** True-positive rates for inner and outer polygon structures and mean Average Precision (mAP) predicted by Encord artificial intelligence.

	20-Image Model	60-Image Model	80-Image Model
Outer Polygon			
mAP	0.85	0.058	0
True-Positive Rate	0.86	0.18	0
Inner Polygon			
mAP	0.29	0.29	0.29
True-Positive Rate	0.23	0.23	0.23

**Table 3 diagnostics-14-00046-t003:** Contingency table of the best-performing model for outer and inner polygons.

	Clinician-Labeled US	
	Accurate	Non-Accurate
Outer Polygon		
Positive (Correct)	19	3
Negative (Incorrect)	3	NA
Inner Polygon		
Positive (Correct)	5	1
Negative (Incorrect)	17	NA

NA stands for does not apply.

**Table 4 diagnostics-14-00046-t004:** The blur metric and number of Doppler images calculated for all training data sets.

	20-Image Model	60-Image Model	80-Image Model	100 Images Total
Blur Metric (Median, IQR)	−138 (-268, 8)	−190 (−367, 19)	−161 (−314, 4)	−183 (−315, 5)
Doppler Images (%, *n*)	35% (7)	33% (20)	33% (26)	35% (35)

## Data Availability

Data that were used in this study are available on request from the corresponding author (T.R.B.). Codes to perform the analyses in this manuscript are available from the authors upon request (T.R.B.).

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
