# Peer review of "Feasibility of Encord Artificial Intelligence Annotation of Arterial Duplex Ultrasound Images"

_diagnostics, 2023, doi:10.3390/diagnostics14010046_

Round 1

Reviewer 1 Report

Comments and Suggestions for Authors

Thank you for giving me the opportunity to review the manuscript.  The study explores the feasibility of using the Encord AI platform for automated segmentation of inner and outer lumens in popliteal artery aneurysm (PAA) duplex ultrasound images, aiming to standardize critical measurements for operative decision-making. 

1) The study relies on a relatively small dataset of 100 images from 44 patients. This sample size might be insufficient to draw robust conclusions, and the generalizability of the findings to a broader population could be questioned.

2) The use of PNG image format instead of the standard Digital Imaging and Communications in Medicine (DICOM) might be a limitation. DICOM is the international standard for medical imaging, and using a lower-quality format could impact the study's applicability to real-world clinical settings.

3) The study did not include normal arteries in the training model. This omission might limit the model's ability to differentiate between normal and pathological conditions. Including normal cases could enhance the model's performance and broaden its clinical utility.

4) The study focuses on the Encord AI platform but does not provide a comparison with other existing AI models for PAA DUS image segmentation. Assessing the performance against established models would provide a more comprehensive understanding of the Encord platform's efficacy.

5) While the study examines the impact of blur on model performance, the evaluation of blur might not be exhaustive. Other factors affecting image quality, such as artifacts or variability in operator skill, could also influence the model but are not explicitly addressed.

6) The selection of images by Registered Physician in Vascular Interpretation (RPVI) certified co-authors might introduce bias. The criteria for image selection and potential subjectivity in this process could impact the study's objectivity.

7) The study reports better performance with a smaller training dataset, which might indicate potential overfitting. Overfit models may perform well on the training set but struggle to generalize to new, unseen data.

Despite promising results in outer lumen segmentation, limitations include the use of lower-quality PNG images and challenges in accurately segmenting the inner lumen, urging caution in fully endorsing the platform's applicability to vascular ultrasound. Reviewers should carefully evaluate these potential pitfalls and consider their impact on the study's validity, reliability, and applicability to real-world clinical scenarios. 

Author Response

We thank the reviewers for their insightful comments. The opportunity to revise our manuscript is greatly appreciated. We believe after appropriately addressing all three reviewer’s concerns, our manuscript is now much improved. The reviewer comments and our bolded responses can be found below.

Specifically, this reviewer #1 has a lot of thoughtful idea on expanded methodologies that we aim to incorporate in our larger future study of proof of concept. Changes to the manuscript text have been tracked according to your instructions. We hope these responses adequately address the reviewer’s concerns. Please do not hesitate to provide more comments if this response is unsatisfactory and thank you for your consideration of revisions for this manuscript.

Reviewer #1:

Thank you for giving me the opportunity to review the manuscript.  The study explores the feasibility of using the Encord AI platform for automated segmentation of inner and outer lumens in popliteal artery aneurysm (PAA) duplex ultrasound images, aiming to standardize critical measurements for operative decision-making.

1) The study relies on a relatively small dataset of 100 images from 44 patients. This sample size might be insufficient to draw robust conclusions, and the generalizability of the findings to a broader population could be questioned.

Response: We completely agree this model was originally created with a very small subset of patients. We opted for a small sample to determine if the use of AI for this purpose was feasible. Certainly we cannot draw robust conclusions concerning Encord and AI. However, if AI were to successfully segment features of PAA that are of clinical interest, they could certainly improve efficiency and eliminate operator subjectivity.

2) The use of PNG image format instead of the standard Digital Imaging and Communications in Medicine (DICOM) might be a limitation. DICOM is the international standard for medical imaging, and using a lower-quality format could impact the study's applicability to real-world clinical settings.

Response: We do acknowledge this imaging limitation in our discussion. Given this is a critical limitation, we have now expanded on this weakness in our limitations section;

“The image format used in this study was PNG, which is a lower quality imaged than Digital Imaging and Communications in Medicine (DICOM) images that are the inter-national standard. The lower quality imaging limits the generalizability of this study. The Encord platform is equipped to handle DICOM storage and future studies should aim to use medical grade DICOM images when training AI models.” Page 10 Lines 360-365

3) The study did not include normal arteries in the training model. This omission might limit the model's ability to differentiate between normal and pathological conditions. Including normal cases could enhance the model's performance and broaden its clinical utility.

Response: We agree that it is important to discern between normal and abnormal arteries. Popliteal artery aneurysms are defined only by size and therefore, developing a tool to measure size will allow us to determine this pathological condition. Limiting this study to popliteal artery aneurysms has allowed to us focus on a homogenous group for accurate AI training. Including normal popliteal arteries in our next pilot or proof of concept study will absolutely help increase the generalizability of this AI model to possibly all vascular ultrasounds.

“We did not include any normal arteries in the training model, which may be easier training material for AI and could be used in future model creation.” Page11 Lines 393-395

4) The study focuses on the Encord AI platform but does not provide a comparison with other existing AI models for PAA DUS image segmentation. Assessing the performance against established models would provide a more comprehensive understanding of the Encord platform's efficacy.

Response: There are currently no established models that we know of for AI segmentation of popliteal arteries. At the time of this study design, we were not sure the Encord platform would be successful in segmenting the arterial wall. Now that we have a proof of concept, we should certainly expand our methods to do a comparative study. We will provide other platforms, such as RedBrick AI and CaptionHealth, the same images and compare the models developed from all three. This would be an informative study to determine advantages and disadvantages of each platform. We have added this addition to our text.

“However, Encord’s generated models could be compared to other AI generated models for accuracy and performance. Providing the same images and comparing the models de-veloped would be an informative study to determine advantages and disadvantages of each platform with regards to vascular ultrasound. Page 11 Lines 397-399

5) While the study examines the impact of blur on model performance, the evaluation of blur might not be exhaustive. Other factors affecting image quality, such as artifacts or variability in operator skill, could also influence the model but are not explicitly addressed.

Response: We agree that blur is not the only component of image quality that can effect model performance. We have now added to our manuscript that we are unable to assess effects of artifact due to our selection bias:

“The smaller datasets had better precision and accuracy than the larger datasets, possibly due to image features and quality. We are unable to quantify image artifact in this study, as we have selected images with minimal artifact. One component of image quality often quantified in AI training is blur” Page 8 Lines 265-269

6) The selection of images by Registered Physician in Vascular Interpretation (RPVI) certified co-authors might introduce bias. The criteria for image selection and potential subjectivity in this process could impact the study's objectivity.

Response: We have introduced bias in our study by selecting high quality images for image training. This was done purposefully to ensure that a model could be adequately trained in this feasibility study, but certainly does limit generalizability. We have now included this as another major limitation:

“In addition, training a model with a large number of both high and lower quality images could improve the generalizability of a model. In this study, the selection of images by RPVI certified physicians introduced bias and decreased the generalizability of this model to all images captured.” Page 10 Lines 372-375

7) The study reports better performance with a smaller training dataset, which might indicate potential overfitting. Overfit models may perform well on the training set but struggle to generalize to new, unseen data.

Response: This is definitely a possibility in this study and we have added this to our limitations:

“We found the best performing model was trained on a small data set of 20 images, which could also represent overfitting of the data that struggles to perform well with new data.” Page 10 Lines 369-370

8) Despite promising results in outer lumen segmentation, limitations include the use of lower-quality PNG images and challenges in accurately segmenting the inner lumen, urging caution in fully endorsing the platform's applicability to vascular ultrasound. Reviewers should carefully evaluate these potential pitfalls and consider their impact on the study's validity, reliability, and applicability to real-world clinical scenarios.

Response: We agree with this review and would like to stress this was only a small feasibility study, aimed at assessing whether or not a model could be created using the images and assessing the model. We absolutely agree with all of the limitations provided by this reviewer and are currently planning for broader studies that can address these limitations in a step by step fashion. Thank you for close review of our manuscript.

Reviewer 2 Report

Comments and Suggestions for Authors

Abstract

I would avoid words such as "tedious" in the introduction of the abstract. What is possibly tedious is highly subjective.

I think the discussion section should be improved.

In fact, the clinical applicability is unclear, nor well discussed.

Why do you think the use of AI in this segment is important? Usually, PAA repair is based upon aneurysm diameter (and not true lumen, which is a minor information), or on symptoms/complications (in an urgency context, an "accurate" US provinding precise measures is of little interest).

How do you think this method can be applied to real world practice? An operator performs a US and sends random images to an online platform?

Will you upload only one image, or an entire set of images?

In your conclusion you state "[...]provide a set of standardized PAA characteristics 310 for clinical decision making". However, this is highly dependent on the images you chose to analyse in the AI platform. Will you provide a protocol for image acquisition?

All these aspects need to be deeply discussed, otherwise your paper is a mere exercise with the encord online AI platform.

Author Response

We thank the reviewers for their insightful comments. The opportunity to revise our manuscript is greatly appreciated. We believe after appropriately addressing all three reviewer’s concerns, our manuscript is now much improved. The reviewer comments and our bolded responses can be found below.

Specifically, this reviewer #2 was concerned about the discussion of this method and the limitations in clinical context. We agree the current methods are not ready for clinical introduction and require more testing and protocoling to be put into real world practice. Changes to the manuscript text have been tracked according to your instructions. We hope these responses adequately address the reviewer’s concerns. Please do not hesitate to provide more comments if this response is unsatisfactory and thank you for your consideration of revisions for this manuscript.

Reviewer #2:

1) I would avoid words such as "tedious" in the introduction of the abstract. What is possibly tedious is highly subjective.

Response:  We do agree with the reviewer that tedious is certainly subjective, so we have replaced the word tedious with “time consuming” in both the abstract and the introduction:

“DUS measurements for popliteal artery aneurysms (PAAs) specifically can be time consuming, error prone, and operator dependent.” Page 1 Lines 11-12

2) I think the discussion section should be improved. In fact, the clinical applicability is unclear, nor well discussed.

Response: The clinical applicability at this phase in this study is limited. However, the clinical potential for application is great and we have therefore added two additional paragraphs in the discussion section explaining our reasoning.

“In terms of overall clinical application, the field of ultrasound (US) has presented sub-stantial opportunities for the integration of AI. Inherent subjective characteristics of US can be improved with the integration of AI, including grayscale imaging quality, adversely affected by operator acquisition [9], and noise in relationship to other structures [10]. The clinical need for segmentation in US has been substantially advanced by AI technology: breast cancer detection [15,16], thyroid nodule classification [17,18], and hepatic tissue vasculature identification in liver US [19]. Specifically within cardiovascular disease, AI has been successfully used to aid with accurate segmentation of the four chambers of the heart [45]. Heart morphology can be affected by disease factors, causing wall thickening, remodeling, and pressure changes that are difficult to manually collect for each image and subjective based on the experience of both the technician and the interpreter. Technology has been developed to automatically segment 2D or 3D images of the heart, where au-tomatic and accurate measurements of cardiac cavity size can be performed. The benefits of automation include not only time but also accuracy: in a convolutional neural network model trained on 14,000 images, automated measurements were either comparable or superior to manual measurements of cardiac chambers and ejection fraction [46].

            The same challenges surmounted by AI within cardiovascular US currently persist in the field of vascular US [47]. The quality of each vascular image is also based on the experience of the technician, and the final report generated is based on the subjectiveness of the interpreter. The Intersocietal Accreditation Commission (IAC) has initiated standards for vascular US acquisition and reports, but not every center is IAC accredited. In addition, the IAC has no current image acquisition or result reporting standards for PAA DUS, and therefore image protocols are left up to the internal protocols of each in-stitution [28,48]. The SVS guidelines focus mainly on quantifying and reporting PAA size [7]. However, size alone does not singularly dictate the need for operative repair; studies have demonstrated that thrombus burden and the percent thrombus also portend a high risk of thromboembolic events and amputation [48-51]. Manual segmentation of the vessel lumen to identify these high-risk features is difficult given the similar echogenicity of adjacent plaques. This similar problem in carotid US has been resolved with the use of AI: machine learning has been applied to the measurement of carotid artery intima-media thickness [52], segmentation of the vascular lumen [53], and classification of carotid vascular plaque components [53]. In this study, we were able to train an existing easy-to-use AI platform on the identification and segmentation of the vascular inner and outer lumen. These measurements can be used to abstract a diameter and percent thrombosis, which are high-risk features of PAA resulting in thromboembolic events [7]. Clinically, applying a model to PAA US has the potential to eliminate measurement subjectivity and provide efficient segmentation for result reporting. The ideal real-world application would include uploading all PAA DUS images to the Encord platform for segmentation and calculation of the largest diameter of PAA and the highest percent thrombus burden within the collection of images. Although the clinical application at this phase in development is severely limited, this study provides a foundation for creation of a more robust AI model.” Pages 9-10 Lines 320-363

3) Why do you think the use of AI in this segment is important? Usually, PAA repair is based upon aneurysm diameter (and not true lumen, which is a minor information), or on symptoms/complications (in an urgency context, an "accurate" US provinding precise measures is of little interest).

Response: Although the focus of clinical practice guidelines have been on size of the total aneurysm, the guidelines also discuss mural thrombus as a clinical criteria for repair. There is new evidence to suggest quantifying thrombus amount could also be proportional to the risk of thromboembolism and therefore acute limb ischemia. Size and thrombus burden quantification can be captured with segmentation of the inner and outer lumen, similar to what has been performed in carotid ultrasounds. We have now elaborated on this point in our discussion:

“The quality of each vascular image is also based on the experience of the technician, and the final report generated is based on the subjectiveness of the interpreter. The Interso-cietal Accreditation Commission (IAC) has initiated standards for vascular US acquisition and reports, but not every center is IAC accredited. In addition, the IAC has no current image acquisition or result reporting standards for PAA DUS, and therefore image protocols are left up to the internal protocols of each institution [28,48]. The SVS guidelines focus mainly on quantifying and reporting PAA size [7]. However, size alone does not singularly dictate the need for operative repair; studies have demonstrated that thrombus burden and the percent thrombus also portend a high risk of thromboembolic events and amputation [48-51].” Page 10 Lines 339-348

4) How do you think this method can be applied to real world practice? An operator performs a US and sends random images to an online platform?

Response: We have not clearly outlines the desired real world application of our model. We have now explained this within the discussion section of our manuscript:

“The ideal real-world application would include uploading all PAA DUS images to the Encord platform for segmentation and calculation of the largest diameter of PAA and the highest percent thrombus burden within the collection of images. Although the clinical application at this phase in development is severely limited, this study provides a foundation for creation of a more robust AI model.” Page 10 Lines 358-363

5) Will you upload only one image, or an entire set of images?’

Response: Providing a set of images per patient would be the most useful, and selecting the largest diameter and largest percent thrombus from these images would help clinicians understand the severity of the disease burden. 

6) In your conclusion you state "[...]provide a set of standardized PAA characteristics 310 for clinical decision making". However, this is highly dependent on the images you chose to analyse in the AI platform. Will you provide a protocol for image acquisition?’

Response: We agree that image acquisition plays a large role in subjective findings on ultrasound. There is an Intersocietal Accreditation Commission that provides guidelines on US image acquisition, but unfortunately there if no protocol for PAA DUS and protocols are left up to the discretion of the institution. We have now more clearly outlined this issue in our manuscript:

“The Intersocietal Accreditation Commission (IAC) has initiated standards for vascular US acquisition and reports, but not every center is IAC accredited. In addition, the IAC has no current image acquisition or result reporting standards for PAA DUS, and therefore image protocols are left up to the internal protocols of each institution [28,48]. The SVS guidelines focus mainly on quantifying and reporting PAA size [7]. However, size alone does not singularly dictate the need for operative repair; studies have demonstrated that thrombus burden and the percent thrombus also portend a high risk of thromboembolic events and amputation [48-51].” Page 10 Lines 341-348

7) All these aspects need to be deeply discussed, otherwise your paper is a mere exercise with the encord online AI platform.

Response: We believe that the major additions to our discussion will help clarify the clinical application of our model and plans for future development. If more discussion is still desired, we would be happy to expand even further.

Reviewer 3 Report

Comments and Suggestions for Authors

An interesting use of AI technology in the vascular lab. Here are my comments and questions:

1)The fact that all datasets came from males, will this interfere with final interpretation of result?

2)The authors attempted to describe features of feasibility of AI technology in Discussion but failed to list these criteria in details? 

3)How is AI differed from current standard of DUS reporting? Additive? Supplementative?

4)One particular features of direction of flow for vascular DUS is that the color changes can may be applied both ways as compared to cardiac echo which is more clearly defined. Will this affect the AI interpretation as suggested in Discussion

5)Authors have suggested PAA lumen segmentation may be affected by vessel wall and adjacent thrombus! How important is it in clinical AI application?

Comments on the Quality of English Language

NO change required

Author Response

We thank the reviewers for their insightful comments. The opportunity to revise our manuscript is greatly appreciated. We believe after appropriately addressing all three reviewer’s concerns, our manuscript is now much improved. The reviewer comments and our bolded responses can be found below. Specifically, this reviewer raised interesting specific questions about images and interpretation.

Changes to the manuscript text have been tracked according to your instructions. We hope these responses adequately address the reviewer’s concerns. Please do not hesitate to provide more comments if this response is unsatisfactory and thank you for your consideration of revisions for this manuscript.

Reviewer #3:

An interesting use of AI technology in the vascular lab. Here are my comments and questions:

1) The fact that all datasets came from males, will this interfere with final interpretation of result?

Response: This reviewer poses a thoughtful question. Although the majority of PAAs occur in males, ultrasound imaging findings of PAA are equivalent in males and females. Notably, there are no disparities in the criteria for PAA aneurysm size between the two genders. Consequently, the inclusion of an all-male cohort is unlikely to influence the result of this study.

2) The authors attempted to describe features of feasibility of AI technology in Discussion but failed to list these criteria in details?

Response: We agree with this reviewer that there is a lack of discussion surrounding the feasibility of the model. We have now added a paragraph discussing the feasibility of the process.

“The generation of the models discussed proved highly feasible from technical, opera-tional, and results perspectives. No specialized or technical training was needed to utilize the Encord platform for model training or testing. We uploaded our ultrasound files to the Encord platform as a data set. We then effortlessly attached this data set to a project. Within the project, we created two ontologies for the inner and outer lumen segmentation. The operationalization of the model training system was seamless, facilitated by the Encord platform’s capability to allow two authors to independently segment each image and verify segmentation within the platform. Discrepancies identified by the platform were promptly flagged for easy identification. Following the annotation and validation of images with the agreed-upon ground truth, model training parameters—specifically, the Mask R-CNN backbone and 500 epochs—were selected within the Encord platform. After training models on subsets of images, testing was also conducted within the Encord platform. The desired tests were simply selected in the analysis tab of the project and, after a runtime period, the platform presented calculations of true positive, false negative, mAP, IoU, and blur. In summary, the platform’s intuitive navigation, complemented by tutorials for both model training and analysis, allowed for straightforward operational-ization of the model training system among members of the research team. The results were displayed in an understandable format and interpreted within the following dis-cussion.”

Page 8 Lines 246-263

3) How is AI differed from current standard of DUS reporting? Additive? Supplementative?

Response: There is an Intersocietal Accreditation Commission that provides guidelines on image acquisition and result reporting for ultrasound. However, these guidelines do not provide protocols for PAA DUS and these systems are left up to the discretion of the institution. We have published a recent study to call for standardization of these protocols by the commission and have now outlined this in our manuscript:

“The Intersocietal Accreditation Commission (IAC) has initiated standards for vascular US acquisition and reports, but not every center is IAC accredited. In addition, the IAC has no current image acquisition or result reporting standards for PAA DUS, and therefore image protocols are left up to the internal protocols of each institution [28,48]. The SVS guidelines focus mainly on quantifying and reporting PAA size [7]. However, size alone does not singularly dictate the need for operative repair; studies have demonstrated that thrombus burden and the percent thrombus also portend a high risk of thromboembolic events and amputation [48-51].” Page 10 Lines 341-348

4) One particular features of direction of flow for vascular DUS is that the color changes can may be applied both ways as compared to cardiac echo which is more clearly defined. Will this affect the AI interpretation as suggested in Discussion

Response: The reviewer introduces an interesting point of discussion. The Encord platform does not include predefined color scales or gradient thresholds. This absence raises the prospect that a model may acquire the ability to identify both blue and red color as an equivalent entity with sufficient training. Our models may have identified both blue and red color to be the inner segmentation of an open lumen, which is a desired interpretation within the context of this study. We have added this hypothesis to our manuscript: 

“Although color doppler directionality changes depending the position of the probe, the Encord platform does not include predefined color scales or gradient thresholds. This absence raises the prospect that a model may acquire the ability to identify both blue and red color as an equivalent entity with sufficient training. Our models may have identified both blue and red color to be the inner segmentation of an open lumen, which is a desired interpretation within the context of this study.” Page 9 Lines 295-300

5) Authors have suggested PAA lumen segmentation may be affected by vessel wall and adjacent thrombus! How important is it in clinical AI application?

Response: There is evidence to suggest thrombus amount could be proportional to the risk of thromboembolism. Thromboembolism is the cause of acute limb ischemia and amputations within this population. Therefore, quantification of adjacent thrombus in PAAs is important for Vascular Surgery and the decision for operative repair. We have now elaborated on the clinical application of currently segmenting thrombus in our discussion:

“The Intersocietal Accreditation Commission (IAC) has initiated standards for vascular US acquisition and reports, but not every center is IAC accredited. In addition, the IAC has no current image acquisition or result reporting standards for PAA DUS, and therefore image protocols are left up to the internal protocols of each institution [28,48]. The SVS guidelines focus mainly on quantifying and reporting PAA size [7]. However, size alone does not singularly dictate the need for operative repair; studies have demonstrated that thrombus burden and the percent thrombus also portend a high risk of thromboembolic events and amputation [48-51]. Manual segmentation of the vessel lumen to identify these high-risk features is difficult given the similar echogenicity of adjacent plaques.” Page 10 Lines 341-350

Round 2

Reviewer 1 Report

Comments and Suggestions for Authors

I acknowledge authors rebuttal.

Reviewer 2 Report

Comments and Suggestions for Authors

Nothing to add.

Thank you